# Next Generation Health Claims Based on Resilience: The Example of Whole-Grain Wheat

**DOI:** 10.3390/nu12102945

**Published:** 2020-09-25

**Authors:** Femke Hoevenaars, Jan-Willem van der Kamp, Willem van den Brink, Suzan Wopereis

**Affiliations:** Research Group Microbiology & Systems Biology, Netherlands Organization for Applied Scientific Research (TNO), 3704HE Zeist, The Netherlands; femke.hoevenaars@tno.nl (F.H.); jan-willem.vanderkamp@tno.nl (J.-W.v.d.K.); willem.vandenbrink@tno.nl (W.v.d.B.)

**Keywords:** whole grain wheat, wholegrain, phenotypic flexibility, biomarkers, challenge test, health claim, resilience

## Abstract

Health claims on foods are a way of informing consumers about the health benefits of a food product. Traditionally, these claims are based on scientific evaluation of markers originating from a pharmacological view on health. About a decade ago, the definition of health has been rephrased to ‘the ability to adapt’ that opened up the possibility for a next generation of health claims based on a new way of quantifying health by evaluating resilience. Here, we would like to introduce an opportunity for future scientific substantiation of health claims on food products by using whole-grain wheat as an example. Characterization of the individual whole wheat grain food product or whole wheat flour would probably be considered as sufficiently characterized by the European Food Safety Authority, while the food category whole grain is not specific enough. Meta-analysis provides the scientific evidence that long-term whole-grain wheat consumption is beneficial for health, although results from single ‘gold standard’ efficacy studies are not always straight forward based on classic measurement methods. Future studies may want to underpin the scientific argumentation that long-term whole grain wheat consumption improves resilience, by evaluating the disruption and rate of a selected panel of blood markers in response to a standardized oral protein glucose lipid tolerance test and aggregated into biomarkers with substantiated physiological benefits, to make a next-generation health claim for whole-grain wheat achievable in the near future.

## 1. Introduction

Health claims on foods are a way of informing consumers about the ‘treats’ of a product. This is an opportunity for the food industry to distinguish their products in the broad market range. In 2006, the European Union approved a set of rules concerning the use of nutrition and health claims for foods (Regulation (EC) No 1924/2006) [1]. One of the main goals of this regulation is to ensure that claims are not misleading for the customer and that regulations are uniform across EU member states for fair market operation and promoting innovation.

The European Food Safety Authority (EFSA) evaluates the scientific basis of claims on nutritional or health benefits of foods. The EFSA only assesses submitted health claims when (a) the food or the food constituent is defined and sufficiently characterized, (b) the claimed effect is based on the essentiality of a nutrient, or the claimed effect is defined and has a beneficial effect on human health, (c) sufficient evidence is present on the cause and effect relationship between consumption of a product and the claimed health effect, and (d) the quantity that is needed to consume the food constituent fits into a normal diet pattern [2]. This is of interest as citizens are becoming more aware and are more actively involved in taking care of their own health. In judging health claim dossiers, the EFSA mainly accepts health claims based on validated and established biomarkers [3]. Currently, validated efficacy biomarkers and measurement methods originate mainly from medical research and focus on showing medical treatment effects on the diseased state. Therefore, most biomarkers which are currently in use and accepted by regulatory authorities are focused on showing effects on disease rather than on health improvement in a healthy range of the population. This complicates the design and execution of science-based intervention studies focusing on the demonstration of health effects in healthy consumers. Recently, the scope on what is a biologically relevant effect (i.e., response of a biological system) has been widened by the EFSA scientific committee, providing opportunities for alternative approaches of demonstrating health effects [4].

Based on the conclusions of an International Invitational Conference on the concept of health, Huber et al. proposed changing the definition of health from the World Health Organization (WHO) formulated in 1948—health is a state of complete physical, mental and social well-being and not merely the absence of disease or infirmity—towards resilience—or “the ability to adapt”—[5,6]. Resilience is the installed capacity of all physiological processes (i.e., metabolism, inflammation, oxidation) to return to homeostatic levels upon a short-term disturbance. Due to ageing, an unhealthy lifestyle or disease, these processes become less flexible. Flexibility (resilience) can be measured by perturbation of homeostasis with a so-called challenge test [7] followed by postprandial measurement of recovery time and amplitude of an array of markers using different analytical platforms [8,9]. A physiological system which is healthier has a better ability to adapt to external stressors than a system with suboptimal health. Here, we would like to introduce an opportunity for future scientific substantiation for health claims of food products by using whole-grain wheat as an example.

## 2. Whole-Grain Wheat and Its Health Effects

Whole-grain wheat intake has been promoted for its beneficial health effects already since the 1900s, when Dr. Thomas Allinson advocated Allinson bread for a healthier lifestyle. In population studies, increased intake of whole grain has been convincingly shown to be associated with a lower risk of a range of chronic diseases and overall mortality [10,11,12,13,14,15,16,17,18,19,20,21]. In most cohort studies (all except for Nordic European populations), wheat is the main source of grains. However, evaluation of individual RCT (randomized controlled trial) intervention studies focused on traditional outcomes (diagnostic biomarkers) provides mixed results. In contrast, a meta-analysis which pooled results from 21 RCTs found that a higher whole-grain intake lowers fasting blood glucose, insulin, total and low density cholesterol, blood pressure, and weight gain [13]. Probably due to the variability in study design, differences in composition of the grains, degree of processing and the definition of the whole-grain intervention provides mixed results in individual RCTs [22]. In 2016, Yamini and Trumbo illustrated this problem by explaining why the relation between whole grains and type 2 diabetes did not lead to a qualified health claim by the US Food and Drug Administration (US FDA) [23]. Of 41 reports of intervention studies reviewed, scientific conclusions could not be drawn from 35 out of 41 studies. Furthermore, for 31 of these studies, the study duration (90 min–12 h) was too short to provide information on long-term health effects. Similar findings are presented for whole-grain consumption and effect on cardiovascular disease risk when limiting evidence only to the strict US FDA definition of whole grains [24,25]. This is also the case for recent evaluation of health benefits on post-prandial glucose regulation [26], lipid regulation [27] and obesity [28]. These opposing results, in combination with weak evidence from classical intervention studies and unclear use of definitions of whole grain, contribute to non-substantiation of health claims, especially when they are not focused on one simple parameter (i.e., cholesterol or glucose reduction) but are aimed at more complex health issues such as cardiovascular disease or type 2 diabetes.

## 3. Chronic Low-Grade Inflammation as a Targetable Example

The six reports identified by *Yamini and Trumbo* as eligible to constitute a qualified health claim for whole grains all unsuccessfully focused on glycemic control [23]. One of the reasons for the absence of beneficial effects may be the fact that other processes precede effects on glucose metabolism. Chronic low-grade inflammation, for example, provides an early mechanistic link between whole-grain effects and cardiometabolic diseases [29,30,31,32,33,34,35]. Indeed, recently, beneficial effects of whole-grain wheat versus refined wheat were shown on inflammation and liver health, but not glucose metabolism [35].

Chronic low-grade inflammation is a condition closely associated with metabolic disorders. Inflammation and metabolism share several signaling pathways, for example the nucleotide-binding oligomerization domain, leucine rich repeat family pyrin domain containing 3 (NLRP3) inflammasome and the c-Jun N-terminal kinase–nuclear factor kappa-light-chain-enhancer of activated B cells (JNK-NFkB) pathway, along which chronic low-grade inflammation is initiated and further developed [30,36,37]. This occurs within metabolic tissues and eventually presents systemically with mildly elevated levels of pro-inflammatory biomarkers (e.g., C-reactive protein, interleukin-6 (IL-6), tumor necrosis factor-alpha) and reduced levels of anti-inflammatory biomarkers (e.g., adiponectin, interleukin-10 (IL-10), transforming growth factor-beta) [32,36]. Given the common signaling pathways, chronic low-grade inflammation can induce several metabolic disturbances within tissues and, systemically, including insulin resistance and atherosclerosis as precursors of diabetes type 2 and cardiovascular disease [31].

Whole grains contain multiple bioactive compounds, including dietary fiber, vitamins, minerals, and antioxidants, that exert beneficial health effects, including anti-inflammatory effects [38]. Individual components or combinations thereof increase the production of short chain fatty acids via microbial fermentation in the colon, positively affect lipid production in and removal from the liver, as beneficial indirect anti-inflammatory effects [32,39]. Additionally, polyphenols (mainly ferulic acid), short chain fatty acids, and other bioactive compounds from whole grains may exert a direct anti-oxidant and anti-inflammatory effect [34,40].

Regardless of the fact that anti-inflammatory effects of whole grains have been studied in RCTs [41,42,43], no qualified health claims have been granted for anti-inflammatory effects of whole grains (Appendix A) [38]. In response to the question how to apply for claims on supporting and maintaining immune function, an EFSA panel stated [44]: “*changes in outcome variable(s) which can be measured* in vivo *in humans by generally accepted methods may not be considered beneficial physiological effects per se if they do not refer to a benefit on a specific function of the body, and thus cannot be the claimed effect (i.e., constitute the only basis for the scientific substantiation of a health claim)”*. The panel explicitly stated that markers of chronic low-grade inflammation do not suffice for the constitution of a health claim. New methods that provide quantitative interpretation for effects on these markers are thus needed.

## 4. Alternative Method for Measurement of Health Effects

As current evidence is accepted for lending credence to the recommendation of incorporating whole grains in general dietary recommendations worldwide [45], it raises the question as to what kind of approach would be useful in the substantiation of health claims. Classically, a biomarker is defined as “a characteristic that is objectively measured and evaluated as an indicator of normal biologic processes, pathogenic processes or pharmacologic responses to a therapeutic intervention” and biomarker evaluation depends mostly on overnight fasting values [46]. Our proposal for new biomarkers is to also take the dynamic nature of biological health processes into account and in line with the rephrased definition of health to ‘the ability to adapt’ [5,6]. This has been taken up into the EFSA guidance document on the assessment of the biological relevance of data in scientific assessment [4].

The first nutritional intervention studies are appearing that show a health impact when evaluating postprandial responses, which is more sensitive than evaluating health in a traditional way by only evaluating overnight fasting values [35,47,48,49,50]. For example, the effect of flavonols in dark chocolate on long-term vascular health, substantiated in EFSA health claim “cocoa flavanols help maintain endothelium-dependent vasodilation, which contributes to normal blood flow” [51] could be reproduced in healthy overweight middle-aged men by evaluating flow mediated dilation (FMD), augmentation index (AIX), total leucocyte counts, and plasma soluble adhesion molecules in response to an oral lipid tolerance test (OLTT) [49].

The next step would be the subsequent integration of these combined data [52], which will yield a quantified degree of resilience of an individual which can be compared to others [50,53,54,55,56,57]. This approach will deliver us a next generation of biomarkers [8] or early biomarkers of disease or effects (Figure 1 adapted from [58]). Based on the results of the Dutch public private partnership PhenFlex and the FP7 EU project Nutritech, a standardized nutritional challenge test (PhenFlex challenge test, PFT) was developed which characterizes how different processes of phenotypic flexibility are being modulated that differentiate between health states in the sequel from optimal health to suboptimal health to diseased [50,57]. The ‘amplitude’ and the ‘duration’ of disturbance (time needed to get back to homeostatic conditions) were quantified for a multitude of biomarkers covering different organs and processes related to metabolic health. This set of biomarkers includes accepted markers by EFSA, scientific well-known markers, not yet accepted by EFSA, and new types of markers generated with modern technologies such as metabolic profiling. This set of biomarkers could be able to empower validity for nutritional health claims by showing the system’s flexibility for optimal health [8].

## 5. Health Claims on Whole-Grain Wheat: Status, Issues and Perspectives

In the European Union, no health claims for whole-grain wheat exist, since no proposals for claims have been submitted. However, a number of claims have been submitted to EFSA for whole-grain foods, diets rich in whole grain and whole grain and whole-grain flour. Most of these claims were focused on either ‘heart health’ or ‘gut health’. Furthermore, beneficial effect upon glucose metabolism, ageing, inflammation/immune and maintenance of a normal body weight are claimed (full overview in Appendix A). None of these claims were evaluated, and rejections were based on”insufficient characterization for a scientific assessment of this claimed effect”. According to the EFSA Panel, whole grains and whole-grain foods were also defined differently across countries.

As stated in Regulation (EC) No 1924/2006, health claims can refer to the health relationship of a food category, a food or to one of its constituents. Whole grains can be considered as a food category. In current definitions of whole grain where the grains are specified, a wide range of grains are included—mostly all grains used for human consumption of the Poaceae family, (including, e.g., wheat, spelt, barley, oats, rye, rice and maize) and often also pseudo-cereals (e.g., amaranth, buckwheat and quinoa) [59,60].

The EFSA Panel concluded for all submitted health claims for food categories that ‘this food is not sufficiently characterized in relation to the claimed effects’. This applies for broad food categories such as ‘vegetable rich diets’, and ‘fruits and vegetables’ [61,62], but also for food categories specified in more detail, as was done for the claims ‘*for peanuts and tree nuts (almonds, hazelnuts, pecans, pistachios and walnuts), excluding brazil, macadamia and cashew nuts)’* [63]. As an exception, the health claim “Meat or fish contributes to the improvement of iron absorption when eaten with other foods containing iron” has been authorized; all types of meat and fish are well characterized for this claim, since they contain at least the required amount of haem-bound iron [64]. Apart from this exception, the EFSA Panel considered only specific products or ingredients as being sufficiently characterized, such as walnuts [65], where a number of the proposed health claims were assessed with favorable outcome.

The well-defined food category, “dietary fiber” (EU, 2008) was also considered by the EFSA Panel as *‘not sufficiently characterized in relation to the claimed effects’*, since these effects can vary depending on the unique physical and chemical characteristics of the specific fiber component [66].

So far, the EFSA Panel considered a wide range of specific fibers as sufficiently characterized, and has substantiated a number of the submitted health claims, including wheat bran fiber for a reduction in intestinal transit time and for an increase in fecal bulk [67], rye fiber for contributing to normal bowel function [68], oat and barley grain fiber for an increase in fecal bulk [69], and beta-glucans from oats and barley for a reduction in post-prandial glycemic responses [70], and for maintenance of normal blood cholesterol levels [70,71]. As is apparent from these examples, the EFSA panel is only considering individual products ‘as sufficiently characterized’; or combinations of ingredients or products if they are similar regarding the component(s) relevant for the claimed effect (e.g., beta-glucans of oats and barley).

### Characterization of Whole Grains, Whole Wheat and Whole Wheat Products

The composition of various whole grains and their flours varies considerably. The level of dietary fiber, an important component in relation to health benefits of whole grains [11], ranges from 4% for rice via 7% (maize), 12% (wheat) and 15% (rye) to 16% for barley (USDA Food composition database). As was the case for dietary fiber, the EFSA Panel will most probably only consider individual whole grains as sufficiently characterized.

The minimum level of whole grain in a product required for calling it a whole-grain product varies considerably between countries, both in- and outside the EU. For example, in Denmark and other northern European countries, at least 50% of the flour in whole-grain bread needs to be whole-grain flour, whereas in many other countries, 100% whole-grain flour is required. It should be noted, however, that the level of whole-grain flour, refined grain flour and all other ingredients needs to be specified in the mandatory list of ingredients.

Contrary to definitions of whole-grain food products, whole grains and whole-grain flour are defined rather uniformly in Europe and worldwide [59]. All definitions describe whole grains with wordings such as in the HEALTHGRAIN definition [59]: “Whole grains shall consist of the intact, ground, cracked or flaked kernel after the removal of inedible parts such as the hull and husk. The principal anatomical components the starchy endosperm, germ and bran are present in the same relative proportions as they exist in the intact kernel.”

In summary, in EFSA assessments of health claims, the characterization of food categories such as fruits and/or vegetables and whole grains is—with one exception—considered as insufficient, whereas the characterization of individual products and ingredients is considered as sufficient. Therefore, we may assume that EFSA will consider whole wheat grains and whole wheat flour as sufficiently characterized for a scientific assessment for health claim substantiation.

## 6. What are the Issues in Substantiation of Health Benefits for Health Claims with Whole-Grain Wheat?

### 6.1. EFSA Requirements for Sufficient Evidence

According to Regulation (EC) No 1924/2006, the use of a health claim shall only be permitted if the food/constituent for which the claim is made has been shown to have a beneficial physiological effect [72]. EFSA has developed guidance documents on the scientific requirements for health claims on six health benefit domains (glucose metabolism and weight management [73], oxidative damage and cardiovascular health [74], bone-joints-skin and oral health [75], neurological and psychological functions [76], physical performance [77], immune system and the gastrointestinal tract and defense against pathogenic microorganisms [78]). These documents contain descriptions of what are considered to be beneficial effects and which outcomes are appropriate for the substantiation of function claims and disease reduction claims. Mid-2014 EFSA indicated that it is necessary to update existing and/or develop new guidance documents for the scientific substantiation of health claims when this is considered appropriate. The guidance documents for claims related to the immune system and gastrointestinal tract, the preparation and presentation of a health claim application, antioxidants, oxidative damage and cardiovascular health have been updated since [2,44,74].

Furthermore, in 2017, a public consultation to receive input from the scientific community and other interested parties was performed on the assessment of biological relevance of data in scientific assessments [79]. It was suggested by the authors that a paragraph should be added to address ‘resilience’ and ‘disturbance of homeostasis’ for evaluation of beneficial health effects from food and nutrition, also from the perspective of health evaluation. The science committee welcomed the proposal and added text in the relevant section addressing the aforementioned definitions. EFSA has processed this in a new guidance document on the assessment of biological relevance [4]. This opens the way for incorporation of new biomarkers with a dynamic character.

A detailed review of the scientific evidence is part of the authorization of a health claim. Both supportive and non-supportive scientific evidence from human intervention studies related to the health claim domain are considered. Depending on the quality and design of the intervention studies submitted, conclusions are drawn in which the hierarchy of evidence is taken into account, as described by the NDA panel.

### 6.2. Substantiation of Scientific Evidence for Whole-Grain Wheat

As mentioned earlier, whole-grain consumption is epidemiologically associated with cardiovascular and metabolic health [18,19,20,80]. The official guidance documents describe a set of traditional biomarkers which are considered valid for the substantiation of a beneficial health effect in this cardiometabolic health arena (Table 1). Unfortunately, the findings of RCT based intervention studies with whole-grain wheat show inconsistent results based upon these traditional biomarkers as compared to the positive reports in population-based studies and meta-analysis [23,41,42,43]. However, the beneficial general health effects of whole-grain wheat consumption are shown by this new ‘measurement of resilience’ approach, as summarized in Table 2 [35]. This RCT was performed with 50 male and female participants that have mildly elevated plasma total cholesterol levels. After a 4 week run-in with a refined wheat intervention, participants were assigned to either a 12 week whole grain (98 g of whole-grain wheat per day) or refined wheat intervention. Before and after the 12 weeks intervention, a PhenFlex challenge test was performed. Blood samples were taken at t = 0, 30, 60, 120, and 240 min for multiparameter analysis focusing on cardiometabolic health, liver health and inflammation. Although in overnight fasting conditions no or little effect of whole-grain wheat was found (only intrahepatic fat content was significantly higher in the refined wheat group [39]), by evaluating resilience it was shown that the 12 weeks whole-grain wheat intervention promotes liver and inflammatory resilience but not metabolic resilience, confirming the effects observed by epidemiological observations. The definition of resilience was based on evaluating the challenge response in a combination of EFSA accepted markers such as blood glucose, insulin, low density lipoprotein (LDL)-cholesterol, high density lipoprotein (HDL)-cholesterol, triglycerides, serum alanine aminotransferase and various interleukins in the context of a reference population [52]. Furthermore, magnetic resonance imaging (MRI) showed that whole-grain wheat consumption prevented the development of a fatty liver [39].

## 7. Conclusions

In order to facilitate future health claim substantiation related to food products from which meta-analyses generate a positive association with beneficial health outcomes, the resilience approach could be a solution. The example of whole-grain wheat teaches us that characterization of one product is preferred above the food category and that whole wheat grains and whole wheat flour probably would be considered as sufficiently characterized by the EFSA. Meta-analysis provides the evidence that whole-grain wheat is beneficial for health although scientific evidence is not always straightforward with classic measurement methods. An alternative measurement approach using a combination of accepted markers showed changes in resilience for whole-grain wheat. Quantification of ‘resilience’ or ‘disturbance of homeostasis’ after a standardized challenge test was accepted by the EFSA scientific committee as a methodology to determine beneficial health effects from food and nutrition. However, the proposed method of measurement of resilience (e.g., differential responses to a standardized oral protein glucose lipid tolerance test as measured by the disruption and rate of response of selected blood markers) needs validation, for example, on long-term clinical relevance. Furthermore, the effect of whole-grain wheat on liver and inflammatory resilience should be confirmed in an independent study showing similar results in addition to showing that changes to the defined response of aggregated markers for liver and inflammatory resilience indeed are related to a beneficial physiological effect. In summary, when these steps to deliver the scientific argumentation that the confirmed observed changes that long-term whole-grain wheat consumption improve liver- and inflammatory resilience, a health claim for whole-grain wheat could be achievable in the near future.

## Figures and Tables

**Figure 1 nutrients-12-02945-f001:**
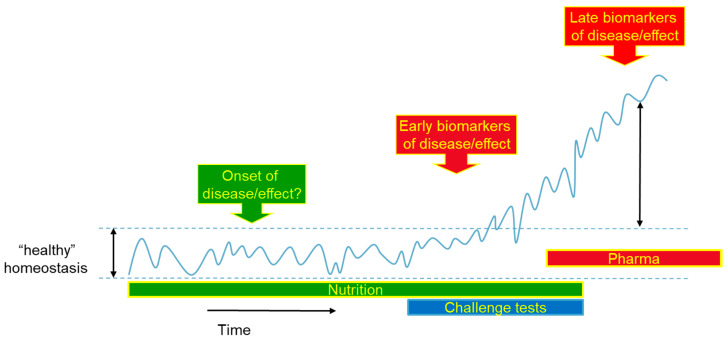
Biomarker sequel from optimal healthy towards disease. In optimal health, a biomarker will show very dynamic behavior within certain normal boundaries. This dynamic behavior is the result of adaptation to daily stressors, such as eating, exercising etc. This dynamic biomarker behavior will start to change in a pre-stage of disease where dynamics of these early biomarkers of disease or effect will just move outside these normal boundaries. When full blown disease develops and progresses, biomarkers will have very different dynamics, as well as different offset values. These so-called late biomarkers of disease or effects are currently mainly in use especially by pharma and health care, which makes sense since they are aiming to show that treatments are beneficial to disease state. However, nutrition is aimed at the maintenance of health or prevention of disease. Reproduced with permission from [58].

**Table 1 nutrients-12-02945-t001:** Traditional biomarkers for health benefits accepted by the European Food Safety Authority (EFSA).

Health Benefit	Biomarker
Nervous system, including psychological functions [76]	Standard psychometric tests, established test batteries or valid and reliable tests for the specific domain. Standard tests of visual acuity and contrast sensitivity or valid clinical diagnostic tools.
Physical performance [77]	Characteristics of the exercise or physical activity in combination with the target population should be specified. Exercise time to fatigue. Outcome measures of muscle function which may be appropriate for the assessment (i.e., change in muscle structure) of the claimed effect in humans in the context of a particular type of exercise or physical activity should be indicated.
Bone, joints, skin and oral health [75]	Measurements of bone mass or bone mineral density using appropriate measures. Maintenance (i.e., reduced loss) of joint function could be assessed via validated protocols and questionnaires. Saliva flow or measurement of self-perceived oral dryness by validated questionnaires. Measurement of trans epidermal water loss using validated methods.
Appetite ratings, weight management, and blood glucose concentrations [73]	Behavioral assessment using methods with appropriate validity and precision. Biochemical markers in support (i.e., cholecystokinin). Body fat (primary; dual energy Xray absorptiometry (DEXA), magnetic resonance imaging MRI, computed tomography (CT), secondary; bodyweight, skinfold thickness, bioelectrical impedance analysis (BIA), air displacement plethysmography (ADP)). Bodyweight regain (prolonged time period, 6 months). Body fat (MRI, CT), waist circumference, sustained effect (12 weeks). Lean mass (DEXA, MRI, CT), specified conditions (physical activity etc.). Blood glucose changes (during time, oral glucose tolerance test (OGTT)). Glycosylated hemoglobin (HbA1c).
Immune system, gastro intestinal, and defense against pathogens [78]	Breath hydrogen levels, gas volume assessed by imaging (i.e., MRI). Transit time, frequency of bowel movements, stool bulk. Validated subjective global symptom severity questionnaires. Composition of the gut microbiota including pathogenic and toxicogenic microorganisms. Changes in immune markers, e.g., numbers of various lymphoid subpopulations in the circulation, changes in markers of inflammation, changes in short chain fatty acid production in the gut, changes in structure of intestinal epithelium, changes in microbiota composition of the gut accompanied by evidence of a beneficial physiological effect or clinical outcome.
Antioxidants, oxidative damage and cardiovascular health [74]	Low Density Lipoprotein-cholesterol Ratio total cholesterol/High Density Lipoprotein HDL-cholesterol Oxidized-LDL Triglycerides Systolic blood pressure Diastolic blood pressure Flow mediated dilatation Decreased platelet aggregation Homocysteine cis-Monounsaturated Fatty Acids

Magnetic resonance imaging (MRI); low density lipoprotein (LDL)-cholesterol.

**Table 2 nutrients-12-02945-t002:** Summary of effects of a refined wheat (RW) versus a whole-grain wheat (WGW) intervention upon health as measured by response to phenotypic flexibility challenge (based upon [35]).

	WGW	RW
Glucose metabolism	~	~
Lipid metabolism	~	~
Liver health	↓	↑↑
Cardiovascular disease markers	O	↑
Inflammatory resilience (based on Interleukin-10, Interleukin-6, Interleukin-8, Tumor necrosis factor-α)	↓↓	↑
Metabolic Resilience (based on glucose, insulin, triglycerides, Low density lipoprotein-cholesterol, High density lipoprotein-cholesterol, total cholesterol)	~	~

Legend; O Possible evidence, no association; ↓ Possible evidence (trend), risk reducing; ↓↓ Probable evidence (significant), risk reducing; ~ Insufficient evidence, no effect; ↑ Possible evidence (trend), risk increasing; ↑↑ Probable evidence (significant), risk increasing.

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
