# Peer review of "Next Generation Health Claims Based on Resilience: The Example of Whole-Grain Wheat"

_nutrients, 2020, doi:10.3390/nu12102945_

Round 1

Reviewer 1 Report

The authors describe an interesting and important topic, namely improvements in substantiation of health claims for foods, with whole grain wheat as example.

The following revisions are suggested:

Article type: this is rather an opinion/viewpoint/commentary type article rather than a systematic review

Abstract: the abstract is very short and uninformative. It could be improved by expanding the results and conclusions of the research. The authors should consider that the abstract is often the only thing that is actually read.

Line 156: Full stop missing before Reproduced.

Line 221: This should be section 6.1?

Line 224: add citations to documents

Throughout: all abbreviations should be defined at first instance

Line 248: This should be section 6.2?

Lines 272-280: Substitute footnotes with reference numbers and include in reference list

Table 2: define WGW/RW in first line

Line 309: Authors contributions are missing

Line 307: Table S1 was unavailable for peer review

Reference list: add DOIs for all references, specifically the many EFSA references to make them easier available

Author Response

The authors describe an interesting and important topic, namely improvements in substantiation of health claims for foods, with whole grain wheat as example.

First of all we would like the reviewer to thank for his/her time and suggestions. Please find below the one-by-one answers to your comments.

The following revisions are suggested:

Article type: this is rather an opinion/viewpoint/commentary type article rather than a systematic review

We agree with the reviewer that the manuscript is more a viewpoint than a systematic review and therefore we have adapted this in line 1 that now states ‘Viewpoint’

Abstract: the abstract is very short and uninformative. It could be improved by expanding the results and conclusions of the research. The authors should consider that the abstract is often the only thing that is actually read.

We agree with the reviewer that the abstract is rather short. Therefore we have expanded the abstract by including our findings on the possibility to achieve a health claim for whole grain wheat. We added the following sentences to the abstract (lines 15-24): “Characterization of the individual whole wheat grain food product or whole wheat flour would probably be considered as sufficiently characterized by the European Food Safety Authority, while the food category whole grain is not specific enough. Meta-analysis provides the scientific evidence that long-term whole-grain wheat consumption is beneficial for health, although results from single ‘gold standard’ efficacy studies are not always straight forward based on classic measurement methods. Future studies may want to underpin the scientific argumentation that long term whole grain wheat consumption improves resilience, by evaluating the disruption and rate of a selected panel of blood markers to a standardized oral protein glucose lipid tolerance test and aggregated into biomarkers with substantiated physiological benefits, to make a next generation health claim for whole grain wheat achievable in the near future.”

Line 156: Full stop missing before Reproduced.

This has been included

Line 221: This should be section 6.1?

This has been adapted

Line 224: add citations to documents

Citations have been added to specify the documents

Throughout: all abbreviations should be defined at first instance

This has been adapted throughout

Line 248: This should be section 6.2?

This has been adapted.

Lines 272-280: Substitute footnotes with reference numbers and include in reference list

All references have been included in the reference list instead of mentioning in the footnotes.

Table 2: define WGW/RW in first line

This has been adapted

Line 309: Authors contributions are missing

Our apologies, part of the explanatory text was still present before the author contributions. This has been removed for clarity on the author contributions.

Line 307: Table S1 was unavailable for peer review

The supplemental table will be uploaded to the portal to make it available.

Reference list: add DOIs for all references, specifically the many EFSA references to make them easier available

The reference list has been adapted to include the DOIs for ease of availability.

Reviewer 2 Report

The manuscript “Next generation health claims based on resilience: the example of whole grain wheat” by Femke Hoevenaars et al. describes the current way to identify and define the health claims for foods, and provides innovative suggestions, in terms of resilience evaluation, to meet the new definition of health as ‘the ability to adapt’. In this context, the authors use whole grain wheat as an example to demonstrate how the measurement of resilience can be used as a novel biomarker for the definition of health claim, although it needs a more specific validation for long-term clinical relevance.

In my opinion, some issues need to be corrected:

1- Supplemental table 1 is missing.

2- Section 4 that describes the concept of resilience should be moved to the introduction section to clarify the concepts that are addressed in the review, with the example of whole grain wheat.

3- Please, define the acronyms “WGW” and “RW” in table 2 legend.

4- Look for typos, grammar, and punctuation errors.

Author Response

First of all we would like the reviewer to thank for his/her time and suggestions. Please find below the one-by-one answers to your comments.

The manuscript “Next generation health claims based on resilience: the example of whole grain wheat” by Femke Hoevenaars et al. describes the current way to identify and define the health claims for foods, and provides innovative suggestions, in terms of resilience evaluation, to meet the new definition of health as ‘the ability to adapt’. In this context, the authors use whole grain wheat as an example to demonstrate how the measurement of resilience can be used as a novel biomarker for the definition of health claim, although it needs a more specific validation for long-term clinical relevance.

In my opinion, some issues need to be corrected:

1- Supplemental table 1 is missing.

Our apologies. The table has now been made available by uploading it into the portal.

2- Section 4 that describes the concept of resilience should be moved to the introduction section to clarify the concepts that are addressed in the review, with the example of whole grain wheat.

Thanks for this suggestion. The paragraph describing the concept of resilience has now been moved to the introduction for full clarification of concepts from the start as suggested by reviewer 2.

3- Please, define the acronyms “WGW” and “RW” in table 2 legend.

This has been adjusted in the figure captions.

4- Look for typos, grammar, and punctuation errors.

We have performed a check on typos, grammar and punctuation errors.

Reviewer 3 Report

Review on manuscript nutrients-927446:

Next generation health claims based on resilience: the example of whole grain wheat

by Femke Hoevenaars, Jan-Willem van der Kamp, Willem van den Brink and Suzan Wopereis

submitted to Nutrients

In the manuscript submitted for comments the Authors introduced an opportunity for future scientific substantiation for health claims of food products by using whole grain wheat as an example.

The topic taken by the authors is current and worth discussing and the manuscript is generally prepared correctly

Detailed recommendation: 

Abstract - should be extended and modified to reflect the content of the manuscript,

lines 146-157 – discussion elements should be removed from the figure caption and transferred to the text,

line 271 – should be: Traditional…,

Table 1 – capital letters should be used consistently,

272-280 – the hyperlink should be removed,

line 283 – “– “ does not appear in the table.

Author Response

First of all we would like the reviewer to thank for his/her time and suggestions. Please find below the one-by-one answers to your comments.

Comments and Suggestions for Authors

Review on manuscript nutrients-927446:

Next generation health claims based on resilience: the example of whole grain wheat

by Femke Hoevenaars, Jan-Willem van der Kamp, Willem van den Brink and Suzan Wopereis

submitted to Nutrients

In the manuscript submitted for comments the Authors introduced an opportunity for future scientific substantiation for health claims of food products by using whole grain wheat as an example.

The topic taken by the authors is current and worth discussing and the manuscript is generally prepared correctly

Detailed recommendation: 

Abstract - should be extended and modified to reflect the content of the manuscript,

We agree with the reviewer and the abstract has been extended to reflect the results and conclusions of the manuscript. We added the following sentences to the abstract (lines 15-24): “Characterization of the individual whole wheat grain food product or whole wheat flour would probably be considered as sufficiently characterized by the European Food Safety Authority, while the food category whole grain is not specific enough. Meta-analysis provides the scientific evidence that long-term whole-grain wheat consumption is beneficial for health, although results from single ‘gold standard’ efficacy studies are not always straight forward based on classic measurement methods. Future studies may want to underpin the scientific argumentation that long term whole grain wheat consumption improves resilience, by evaluating the disruption and rate of a selected panel of blood markers to a standardized oral protein glucose lipid tolerance test and aggregated into biomarkers with substantiated physiological benefits, to make a next generation health claim for whole grain wheat achievable in the near future.”

lines 146-157 – discussion elements should be removed from the figure caption and transferred to the text,

This element has been removed from the figure caption. As the text is already present in the manuscript in Line 151-153 this has not been transferred to the text of the manuscript.

line 271 – should be: Traditional…,

This has been adapted

Table 1 – capital letters should be used consistently,

This has been adapted

272-280 – the hyperlink should be removed,

The hyperlink has been removed and references have been taken up into the reference list.

line 283 – “– “ does not appear in the table.

This has been removed from the table